# Bridging the Gap: Analyzing the Relationship between Environmental Justice Awareness on Twitter and Socio-Environmental Factors Using Remote Sensing and Big Data

Charles Knoble and Danlin Yu *

Department of Earth and Environmental Studies, Montclair State University, Montclair, NJ 07043, USA;
knoblec1@montclair.edu
* Correspondence: yud@montclair.edu; Tel.: +1-973-655-4313

**Abstract:** Mounting awareness of the discriminatory distribution of environmental factors has increasingly placed environmental justice at the forefront of discussions on sustainable development, but responses to these disparities are often too little, too late. Remote sensing has emerged as a potential solution to this problem, capitalizing on the ability to capture high-resolution, spatially explicit data in near-real time. However, a conventional reliance on physical measurements and surface-level analyses risks overlooking the experiences and perceptions of affected communities. It is against this backdrop that the potential integration of remote sensing imagery and socially sensed big data such as social media data assumes a novel and promising role. This study aims to discern the feasibility, opportunities, and implications of integrating the spatial insights provided by remote sensing with the experiential narratives shared on social media platforms, bridging the gap between objective environmental data and community-driven perspectives. We explore this subject in two ways, analyzing the geographic relationship between environmental justice Tweets and environmental justice factors, and reviewing Tweets produced during an extensive wildfire. Remote sensing indexes for green and blue space were reviewed and tested, selecting the measures of best fit to act as independent variables alongside traditional environmental justice factors in the broader analysis. Results from regression models indicate a negative relationship between the number of Tweets utilizing environmental justice relevant terms and the presence of ecosystem services as captured by an NDMI, suggesting a broad awareness of injustice and a relationship between remote sensing and social media. However, there is simultaneously a negative relationship between socially vulnerable populations and Tweets with environmental justice words. This suggests that generally, there is discussion on Twitter about injustice when resources are not present, but the voices of vulnerable populations are often less visible, either as a result of urban bias or a lack of concern for injustices due to habitual ignorance. Our study demonstrates the potential for integrating remote sensing imagery and social sensing data to play a substantial role in detecting injustices and corroborating data collected through community science initiatives.

**Keywords:** environmental justice; Twitter; spatial big data; urban remote sensing; social sensing; New Jersey

## 1. Introduction

In a world grappling with pressing environmental challenges, the issue of environmental injustice has garnered heightened attention. Environmental injustice, often manifested as disproportionate exposure to environmental hazards or lack of access to ecosystem services for socially vulnerable communities [1,2], underscores the urgent need to identify and rectify disparities in access to a clean and healthy environment. However, despite ongoing efforts to address environmental injustices, biased institutions and historic exclusionary

practices continue to perpetuate a climate of environmental inequity across the globe [1–4]. Moreover, conventional tools for detecting these injustices exhibit limitations, falling short of providing a comprehensive and timely understanding of disparities. These challenges stem from a reliance on traditional data sources and methodologies, which may fail to capture the intricate spatial and social dynamics that contribute to environmental inequalities in a timely manner [5]. In the United States, federal mapping tools like EJScreen [6] and state tools like New Jersey's EJMAP [7] draw on infrequently collected Census and environmental datasets to assess current environmental injustice levels. By relying on outdated data, these tools risk leading policymakers to misallocate program resources that could otherwise be directed to mitigate emerging inequities.

As an example, the city of Newark, New Jersey is well known as a community of color experiencing environmental injustice, due in large part to historical discrimination and the prevalence of hazardous facilities [8]. The data required to identify this city as one experiencing environmental injustice are generated by state and federal entities who monitor known hazards. However, if a new hazard were to arise, community members would be required to attract the attention of these governing entities before action would be taken. In the case of Newark, lead contamination in drinking water was emerging as a concern for residents, but remediation actions were not taken until enough data were collected and shared, clearly identifying the problem [9]. Simultaneously, data would not be collected by state or federal agencies until there was a clear reason to investigate. Even after data collection began, testing for lead contamination is a notoriously resource-intensive and geographically narrow process [10], leading to a slow response that prolongs exposure. As a result, a reliance on traditional static measures for environmental injustice creates a false sense of security at a high level, despite the growing presence of an environmental hazard. This study aims to address these shortcomings by investigating the integration of remote sensing and social sensing big data, allowing for the creation of more relevant, farther-reaching environmental injustice detection tools.

In seeking to develop a tool centered around environmental justice, we must first establish a foundational understanding of the subject. In practice, environmental justice is often measured by the levels of injustice present, defined by a disproportionate exposure of environmental burdens or hinderance of access to ecosystem services for socially vulnerable populations [11–15]. Social vulnerability refers to the negative impact sociocultural identity, socioeconomic status, and physical capabilities can have on an individual or community's ability to respond to perturbations, often as a result of historic marginalization [16]. When considering the economic, social, and political resources individuals rely on when preparing for and responding to catastrophic events such as natural disasters, it is clear that those who historically received less resources or were actively encumbered are at a further disadvantage from the outset [16–18]. This phenomenon and environmental justice broadly are relevant at both the individual and community level [16,19].

To identify and measure environmental injustice, researchers traditionally draw upon existing datasets collected by outside sources or set out to collect new data. On the one hand, relying on existing datasets provides a larger quantity of information to be collected over a greater area for a lower price point, often at the cost of temporal relevance [20–24]. On the other hand, producing new datasets allows researchers to control the frequency, methodology, and overarching intention of data collection given they have the resources required to commit to collection efforts. Collecting new datasets also offers an additional opportunity to explore more complex environmental justice topics, such as recognition and representation justice [2,13,25–27]. These data collection limitations are universal in research, but are particularly salient in the realm of environmental justice, as delayed identification of injustices may result in prolonged suffering.

In this context, the field of remote sensing has emerged as a powerful ally in environmental justice research. The ability of remote sensing technology to capture high-resolution, spatially explicit data in near-real time has significantly enhanced our capacity to map environmental hazards and resources and assess their distribution across landscapes. This

technology has bolstered the identification of potential hotspots of environmental injustice, shedding light on areas burdened by air pollution [28,29], natural disasters [30], or compromised ecosystem services [3], particularly through measures like the normalized difference in vegetation index (NDVI) [21,27,31,32]. This application is well conveyed in a review by Kshetri et al. [33], examining four academic publications that leveraged remote sensing imagery to measure deforestation in the Global South. In all four cases, the authors used satellite imagery to trace deforestation rates over time and compared results to estimates made by the entities responsible for the damage. Each case study showed that vegetation loss was greater than what was expressed, leading to legal cases against the responsible party which ultimately resulted in reparations. In this way, remote sensing played a pivotal role in facilitating environmental justice. Similarly, authors Kolosna and Spurlock [3] used remote sensing imagery to compare the distribution of urban tree cover and socially vulnerable communities, with particular attention on socio-political decision-making mechanisms. Results from this analysis showed a clear inequitable distribution of urban tree cover that, despite the prevalence of tree maintenance ordinances, was not acknowledged by local policy mechanisms.

However, while remote sensing has proven invaluable, existing approaches often lack the depth necessary to fully capture the complexities of environmental injustice. The conventional reliance on physical measurements and surface-level analyses risks overlooking the human experiences and perceptions of affected communities, inadvertently neglecting the socio-economic factors that intertwine with environmental disparities. It is against this backdrop that the potential integration of remote sensing imagery and big data such as social media assumes a novel and promising role. By integrating the spatial insights provided by remote sensing with the experiential narratives shared on social media platforms, an unprecedented opportunity arises to bridge the gap between objective environmental data and community-driven perspectives.

This emergence of social media platforms, particularly as a data source for researchers, is a relatively recent phenomenon which has created a new hub for socially sensed big data [34,35]. This term, big data, refers to large quantities of information that are often produced quickly and consistently over large geographic extents [36]. While still in its early stages of development, scholars have begun to refer to big data like social media as a new form of remote sensing data, labeled social sensing data, with individuals serving as the ultimate "sensors" [37–39]. These socially sensed datasets have been utilized for environmental justice investigations before, drawing from pictures posted online to social media [22], locations associated with cellphone use [36], and general social media application activity [40–42]. Authors Xu, Jiang, Li, Zhang, Zhao, Abbar, and González [36], for example, analyzed exposure to particulate matter by incorporating cellphone data into traditional pollution modelling strategies to bolster analytical capabilities. In this case, location information allowed the authors to infer individual home, work, and commute information, in turn providing a foundation on which to build a model for vehicle usage and subsequent particulate matter emissions.

Independently, remotely sensed information such as satellite imagery and socially sensed data can be used to establish a foundation for environmental justice investigations. However, the amalgamation of these datasets offers the potential to not only pinpoint environmental factors, but also to capture the human dimensions of vulnerability and resilience. This integrative approach, while a departure from traditional methods, carries the potential to inform a more comprehensive and robust understanding of environmental injustice in near-real time if necessary. By leveraging this interaction, this approach seeks to open the door to a future in which we may tap into the collective wisdom of communities, amplify their voices, and uncover hidden dimensions of environmental disparities that could reshape our strategies for equitable policy formulation and advocacy.

Previous research has leveraged the synergistic relationship between remote sensing imagery and socially sensed big data before. Wang et al., for example, used the interactions between imagery and socially sensed data to interpolate missing information in their

poverty analysis [43]. However, to the best of our knowledge, this intersection is yet to be adequately explored in the field of environmental justice. In recognition of this untapped potential, this study aims to discern the feasibility, opportunities, and implications of applying remote sensing imagery and social media data together in an environmental injustice investigation. More specifically, we seek to answer the following research questions: (1) Is there a discernible relationship between environmental justice factors inferred from remotely sensed earth observations, traditional governmental sources, and social sensors? (2) How can we best model this relationship using Twitter data, social vulnerability measures, and environmental factors? (3) What are the potential consequences of leaning on these remote sensing datasets to draw conclusions about environmental justice?

We hypothesize that a lack of ecosystem services and prevalence of socially vulnerable populations and environmental hazards will have a positive relationship with Tweets using environmental justice terms, demonstrating an awareness of injustices. To test this hypothesis, we conduct our analysis in two parts. First, we conduct a broader analysis, focused on building a model in which socially sensed environmental justice can be compared to environmental factors, including data drawn from remotely sensed imagery, and social vulnerability. We analyze the geographic relationship between frequency of Tweets with environmental justice terms and environmental justice factors at the Block Group and Tract level using Ordinary-Least Squares regressions and Spatial Autoregressive Models. Independent variables for blue space and green space are drawn from Landsat-8 imagery following an index selection process, allowing us to select the measure which best fits the model. We calculate our dependent variables as the frequency with which geolocated Tweets with environmental justice words appeared within each Block Group and Tract. These final Tweet counts are weighted by area and normalized using the Inverse Normal Transformed technique [44].

Second, we further characterize the relationship between socially sensed big data and environmental factors by investigating Tweets during the state's largest wildfire of the year. MODIS imagery is used to define the extent of wildfire smoke exposure within cloud structures and compare Tweets falling within its boundary to those outside. Air quality data and local news reports are then used to corroborate the extent and impact of smoke exposure. Finally, Tweets are reviewed for all three days of the fire to determine if, and to what extent, the topic of air quality or smoke is discussed each day.

The remainder of this article is broken into the following parts: Section 2 outlines the data sources, methodologies, and the overall research process used in this study. Section 3 outlines the statistical results of the analyses before Section 4 draws conclusions based on the findings. We close with a further discussion surrounding the implication of the study and its limitations.

## 2. Materials and Methods

### 2.1. Research Area

This study focuses on eight of New Jersey's northern counties, ranging from Sussex County in the northwest to Union County in the southeast (Figure 1). This region was selected due to its wide range of urban development, sociodemographic makeup, and environmental characteristics, culminating in a diverse environmental justice landscape. As the closest region to neighboring New York City, New Jersey ranks as one of the most densely populated states in the US, with a high rate of ethnic and religious diversity to match [45]. The state's transition away from industrial production decades ago left behind a legacy of environmental hazards in the form of brownfields, landfills, and contaminated waterways, the likes of which are still being rehabilitated today [46]. These burdens are particularly prevalent in the northern part of the state and, when paired with the echoes of discriminatory policies impacting resource distribution, result in an increased awareness of the persistence of environmental racism and environmental injustice. In fact, the state of New Jersey has openly acknowledged environmental injustice and has begun taking steps toward addressing it, exemplified by legislation passed to mitigate the development

of construction facilities in overburdened communities [47] and address existing pollutants focused in low-income and communities of color [48]. This growing public interest in environmental justice coupled with the state's history and high levels of data availability makes New Jersey an ideal location for conducting novel environmental justice research.

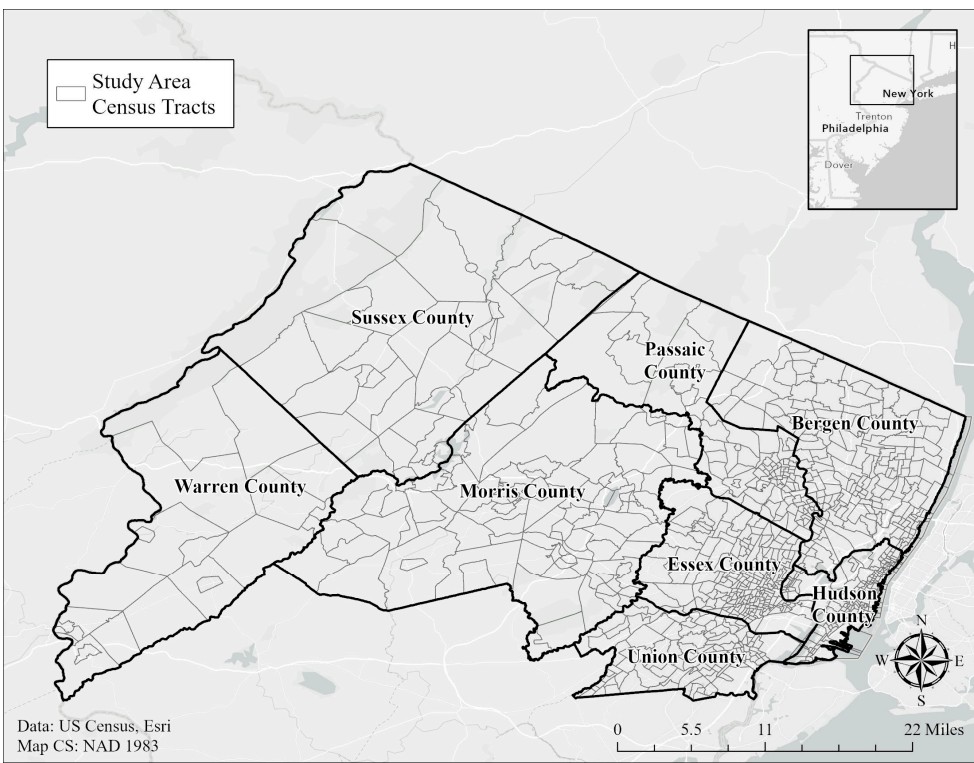

**Figure 1.** Northern New Jersey Counties Selected as Study Area.

In choosing an extent and resolution for our study, the socioeconomic and demographic data drawn imposed limitations. The Census Block Group and Census Tract levels were chosen as the level of analysis in an attempt to acknowledge and mitigate the impact of the Modifiable Areal Unit Problem (MAUP) through comparison [49]. Additionally, we utilize Census boundaries to more directly attribute results to the city or township in which they were identified. We hope that this in turn provides an opportunity to encourage local political action which may promote environmental justice in spite of data aggregation biases [49–52].

### 2.2. Data Collection and Pre-Processing

This study relied on four different types of data: social sensing data, remotely sensed imagery, additional environmental factors, and social vulnerability. The following sections detail each of these.

### 2.2.1. Social Sensing Data Derived from Twitter

Twitter (renamed "X" in 2023) is a micro-blog social media platform that allows individuals to post images, videos, and text up to 280 characters. The platform is used worldwide by individuals and companies alike, boasting 126 million daily active users in 2019 alone [53]. For the purpose of this study, we collected original posts made on Twitter (called 'Tweets') in 2019 from within 25 miles of the study area's center. These Tweets were collected from the Twitter API using R's *rtweet* and *academictwitteR* packages [54,55]. Our query collected original content produced by an individual, removing quotes, retweets, promoted posts, replies, verified posts, and Tweets not written in English. This produced over 6 million total Tweets in and around our study area, only 2,273,667 of which contained

geographic coordinates. Further data cleaning resulted in the removal of duplicate Tweets (posts of the same content from the same location) and Tweets from two accounts that only shared traffic updates. These Tweets were reduced to the study area, resulting in a sample of 270,245 total Tweets from 21,950 unique authors.

For the purpose of the wildfire case study, Tweets were preserved at this stage to ensure full textual content was maintained. Tweets produced during the wildfire containing the words *fire*, *smoke*, or *air* were extracted and reviewed by hand to contextualize discussions. For the broader exploration of environmental justice Tweets throughout the year, the data were further limited to only include Tweets that used words relating to environmental justice. To select which words would fit this criterion, we conducted a semantics analysis of abstracts from environmental justice publications available in the Web of Science database as of 2 December 2022. While relying on academic articles to create a set of environmental justice keywords is clearly biased to the subject in an academic context, this approach offers a more objective route for producing keywords that simultaneously confines terms to the realm of environmental justice. In addition, we also hope the sweeping number of topics addressed across studies will offer a sufficient range of terms that capture the most important keywords.

Searching the term, "environmental justice" in the Web of Science database produced 14,844 results. These publications were downloaded, cleaned to remove duplicates, and scraped using R to capture all words appearing in abstracts. The frequency with which each word was used across publications was calculated and similar words (such as environment and environmental) were combined where necessary. This resulting word list was then filtered to only include words appearing at least 1000 times across abstracts. We then reviewed the content of a subsample of nearly 80,000 Tweets to identify instances in which words from this list of environmental justice terms appeared. This process achieved two goals: first ensuring these terms appeared at least once in our sample of Tweets, and second that appearances of the words were accurately captured beyond the use of the words by themselves. Searching for the 42 terms in this way returned 77 total words, compound words, and hashtags from our sample of Tweets. These words are listed in Table 1 and act as our finalized list of environmental justice terms.

**Table 1.** Environmental Justice words selected from abstract search.

| List of Environmental Justice Terms | | | | | |
|---|---|---|---|---|---|
| access | eatlocalgrown | groups | lakecommunity | pollution | underwater |
| air | economic | hardwork | land | prework | urban |
| climate | energy | health | local | public | veganfood |
| community | environment | healthier | locally | quality | water |
| communityoutreach | environmental | healthy | localmusic | risk | waterfall |
| communityservice | food | healthyfood | management | seafood | waterfront |
| development | foodie | healthymeals | mgfoodsafety | shoplocal | waterside |
| difference | foodphotography | human | njfood | social | work |
| different | foodsafetyisaboutpeople | impact | njfoodie | socialize | worker |
| differentandable | foodsofinstagram | impactinglives | nofarmsnofood | stillwater | working |
| differently | gatedcommunity | issue | people | superfoods | workingdjs |
| dowork | global | issues | poetrycommunity | sustainability | writingcommunity |
| drinklocal | green | justice | policy | system | |

It is important to note that this process intentionally casts a wide net when defining environmental justice Tweets. While it is possible to manually or algorithmically review and sort Tweets to ensure the appearance of these keywords relates to a defined context of environmental justice, we choose instead to trust the more objective nature of keyword frequencies. This is because we hope to capture both recognizable and latent relationships between factors in a manner similar to machine learning algorithms, which analyze big data by assuming connections between factors based on patterns and statistical reporting. These relationships do not have to make sense, and often do not, but nonetheless can be used

to create effective inference tools. Machine learning techniques as a whole are becoming more popular across sectors, from the medical field to security [56], but applying these methods in social and environmental sciences can introduce unintended consequences. More specifically, leveraging machine learning in the context of environmental and social sciences without first conducting a thorough exploratory analysis risks perpetuating human and institutional biases. For this reason, we seek to conduct our analysis in a manner that mitigates the likelihood that relevant relationships are wrongfully dismissed due to unfamiliar or unrecognized connections, while simultaneously operating at a level at which institutional bias can be drawn out and scrutinized.

With this in mind, we use this list to reduce the sample of Tweets to include only those which contained at least one environmental justice term. This process left 15,090 Tweets produced by 4059 authors. These data were plotted as points in GIS and aggregated to their respective Block Group and Tract to establish the total number of environmental justice Tweets appearing within each. In both cases, areas that failed to capture any environmental justice Tweets are assigned a Tweet count of zero in order to be included in our analysis. Finally, Tweet counts are normalized by area of their associated Tract or Block Group.

### 2.2.2. Remotely Sensed Imagery
Green and Blue Space Observed Remotely

We further capitalized on remote sensing techniques to derive green and blue space independent variable measures utilizing Landsat 8 imagery. Low cloud cover imagery was collected for 22 September 2019 at a resolution of 30 m and analyzed in ArcGIS Pro. This image was selected due to the clear atmospheric conditions and the fact that this date falls in the midst of the growing season in New Jersey, which is generally between late March and mid-October. Two images were mosaiced to establish a single layer for analysis.

To ensure we create a model that best expresses the relationship between environmental information drawn from remote sensing imagery and environmental justice experience derived from social media data, we generate and test raster layers from several different spectral indexes. Spectral indexes, or quantitative measures derived from the reflectance values of specific spectral bands in remote sensing data, aim to capture and quantify certain characteristics or properties of the Earth's surface [57]. The NDVI is an example of a vegetation index popularly used in environmental justice research [21,27,31,32]. NDVIs leverage remote sensing technology to measure and compare wavelengths of light sensitive to the chlorophyll in vegetation. In this way, NDVIs quantify vegetation greenness as a representation of plant density by comparing light reflected in the near-infrared spectrum and light reflected in the red range of spectrum.

However, like all spectral indexes, the NDVI is not a perfect measure for vegetation. For one, the measure is less accurate in areas with particularly high density of vegetation. This is because, as vegetation density increases, near-infrared light reflected by the vegetation keeps pace, while reflected visible red light starts to decrease. This means the NDVI becomes oversaturated as red light fails to be accurately measured, making the measure less sensitive to changes in vegetation. Additionally, as with all remotely sensed imagery, the NDVI is impacted by shortcomings associated with capturing images broadly [58]. Factors like shadows and the angle of the sun impact the way that visible light is captured, introducing potential bias that may lead to an underestimation of vegetation.

Other vegetation indexes such as the Modified Soil Adjusted Vegetation Index (MSAVI) may act as an alternative, but the size of the study area may introduce a level of variability in atmospheric qualities, soil characteristics, and vegetation type that would present additional complications [59]. For example, indexes like the MSAVI are suited for pointed investigations in which measures would benefit from controlling soil reflectance. However, in areas with disproportionately high levels of exposed soil, MSAVI is still heavily skewed due to their reliance on the red band. As such, applying these measures across large areas nonetheless risks generating biased results due to the high variability in soil properties and exposure levels. Along this same line, the size of our study area introduces high variability

in vegetation type, vibrancy, and density. Whereas the NDVI is generally highly sensitive to variations in vegetation characteristics, the MSAVI may not be able to provide the same level of robust, consistent results. With these wide-ranging benefits and shortcomings applying to numerous spectral indexes, we test a small selection of vegetation indexes to choose one that most appropriately fits our model. These measures and their equations are listed in Table 2 and include the NDVI, MSAVI, VARI, and NDMI. The linear models' R-square and Akaike information criterion (AIC) values will be used to select the index which best fits the model.

The Landsat imagery was also used to investigate the distribution of ecosystem services in the form of blue space in our study area. Similar to greenery, the extent of water bodies can be estimated using surface reflectance. One example of a water body index is the Modified Normalized Difference Water Index (MNDWI), ref. [60] an adaptation of the Normalized Difference Water Index. The MNDWI incorporates modifications to enhance its performance, reducing the impact of bias introduced by built-up features. It uses green and short-wave infrared bands to identify open water areas such as lakes, rivers, and reservoirs. The MNDWI is based on the principle that water bodies exhibit high reflectance in the green band due to the absorption of chlorophyll, while they have low reflectance in the short-wave infrared band due to the absorption of water itself. By calculating the normalized difference between these two bands, the MNDWI enhances the contrast between water and non-water features, making it easier to identify water bodies.

The MNDWI is, of course, not without its limitations. As the index simply relies on the green and short-wave infrared bands, certain non-water features can be captured in a similar manner to water features due to their reflectance values. On the flip side, the MNDWI can struggle to detect shallower water bodies, as their reflectance properties may resemble surrounding features. Built up areas and vegetation are particularly impactful, displaying higher levels of reflectance that can be mistaken for water bodies. As such, we test two water body indexes, the MNDWI and the AWEI, and proceed with the measure which best fits our model.

**Table 2.** Vegetation and water spectral indexes.

| Index | Formula | Citation |
|---|---|---|
| Normalized Difference Vegetation Index (NDVI) | $\frac{NIR-R}{NIR+R}$ | Kriegler, 1969 [61] |
| Modified Soil Adjusted Vegetation Index (MSAVI) | $\frac{2 \times NIR+1-\sqrt{(2 \times NIR+1)^2-8 \times (NIR-R)}}{2}$ | Qi et al., 1994 [62] |
| Visible Atmospherically Resistant Index (VARI) | $\frac{G-R}{G+R-B}$ | Gitelson et al., 2002 [63] |
| Normalized Difference Moisture Index (NDMI) | $\frac{NIR-SWIR1}{NIR+SWIR1}$ | Gao, 1996 [64] |
| Modified Normalized Difference Water Index (MNDWI) | $\frac{G-SWIR}{G+SWIR}$ | Xu, 2006 [60] |
| Automated Water Extraction Index (AWEI) | $4 \times (G - SWIR2) - (0.25 \times NIR + 2.75 \times SWIR1)$ | Feyisa et al., 2014 [65] |

Note: *NIR* is near-infrared band pixel values, *R* is red band pixel values, *G* is green band pixel values, *B* is blue band pixel values, *SWIR*1 is short-wave infrared band 1 pixel values, and *SWIR*2 is short-wave infrared band 2 pixel values.

Estimating Smoke Distribution Using Imagery

To better contextualize the broader study and explore the potential benefits and consequences of combining remote sensing data, we examine Tweets produced during a natural disaster. We will draw out specific content produced in this time for further scrutiny so that we may then more accurately identify the nature and sentiment of Tweets and determine whether these characteristics vary geographically. Bearing in mind our hypothesis that a higher number of environmental justice Tweets will be present in areas with disproportionate levels of social vulnerability and hazards, we anticipate a higher number of Tweets discussing the environment will be present in areas experiencing the densest smoke coverage. If the results from our broader analysis and pointed case study

facilitate similar understandings, we may conclude that a synergistic relationship is present between both types of remote sensing data collected in the context of environmental justice.

For this case study, we review the content of Tweets produced in areas experiencing the greatest levels of smoke from a local wildfire, using remote sensing imagery to estimate smoke distribution. We investigate the Spring Hill Wildfire in particular, which began the afternoon of 30 March 2019 and was contained 1 April. The fire reportedly burned for approximately one week total, consuming 11,683 acres of land [66,67]. The magnitude of this disaster meant that, although the fire itself was in southern New Jersey, the smoke travelled far beyond.

We draw remote sensing imagery from within the fire's three-day period leading up to containment to visually identify and mark the approximate boundary of the visible smoke patterns. We have selected NASA's Moderate Resolution Imaging Spectroradiometer (MODIS) MOD09GA v006 product for this investigation, providing Terra surface reflectance data at a resolution of 500 m for bands one through seven. Imagery produced by MODIS is particularly well-suited for this purpose due to its frequent visits, often offering daily data products. Data was downloaded for 31 March 2019, capturing imagery from the peak of the fire.

Relying on remote sensing imagery to investigate smoke presents unique intricacies which must be accounted for in order to draw meaningful conclusions. First, the short window of time associated with wildfires means investigations are highly time-sensitive and require the researcher to capitalize on whatever data is available. In this case, the height of the Spring Hill Wildfire happened to be a particularly cloudy day, obscuring the ground and making it difficult to distinguish between clouds and smoke. However, as seen in research by Zhao, et al. [68], the right combination of short-wave infrared and near-infrared spectral bands can help to highlight smoke within and among cloud structures.

Second, the exact extent and impact of smoke is difficult to discern when relying solely on imagery. Imagery does not provide information regarding the altitude at which the smoke was detected or the overall vertical composition of the air column. This means that the smoke identified may be present low in the atmosphere, but not necessarily at the ground level. In the case of our study, this might alter the discussions occurring on Twitter. If smoke is present on the ground, individuals would be more likely to Tweet about their experience interacting with the smoke, such as the impact of poor air quality on their health or the smell of smoke in the air. Alternatively, if smoke is only present at higher altitudes, the impact would likely be less severe. This might lead individuals to Tweet less frequently and about different aspects of the experience, such as the sight of the smoke or news articles on the subject. Despite this potential variation in severity, we expect that including the words *smoke* and *fire* alongside the keyword *air* will nonetheless capture discussions on environmental conditions.

Third, the nature of wildfires and smoke are such that they are continuously changing, whereas remote sensing imagery is a snapshot capturing information at one point in time. Although our data show the distribution of smoke at the moment the image was captured, the smoke continued to change after it was taken, making it difficult to draw sweeping conclusions regarding its distribution throughout the day. We seek to address these concerns through our use of ground-based observations. $PM_{2.5}$ data are collected for EPA monitoring stations on 31 March and interpolated using the same methods described in Section 2.2.3. This layer is then overlayed with the smoke extent identified in the MODIS imagery for validation. Furthermore, news articles discussing the wildfire are reviewed to identify the impacts described by observers. Each of these datasets will bolster the results of the MODIS investigation by comparing the visually identified smoke extent to air quality on the ground, as described by air monitors, news reports, and Twitter users.

### 2.2.3. Additional Environmental Factors

Additional environmental data were included in the model to characterize the relationship between environmental justice Tweets and relevant factors. Potential variables were

identified and plotted to test assumptions of linearity. Those which met this requirement were grouped and further checked for covariation before a final set was chosen. Environmental characteristics that were examined but deemed unfit for the model included groundwater contamination, brownfields, parks, and landfills.

The selected environmental factors are listed in Table 3 and broken into two categories, hazards, and resources. Hazard variables include things that could directly facilitate harm, result in damage, or cause some level of discomfort. We use contaminated sites, flood zones, and Particulate Matter 2.5 ($PM_{2.5}$). Similarly, environmental resource data included factors relating to ecosystem services, sustainability investment, or facilitating access to natural spaces. We utilize green spaces (greenery), blue spaces (waterbodies), alternative fueled vehicle (AFV) fueling stations, and transit stations for these purposes.

**Table 3.** Hazard and resource measures (independent variables).

| Variable | Measure Used for Each Block Group or Tract |
| --- | --- |
| Contaminated Sites | Distance to Nearest |
| Flood Zones | Percentage of Total Area |
| $PM_{2.5}$ | Mean Annual Concentration ($\mu g/m^3$ LC) |
| Green Space | Mean Value |
| Blue Space | Mean Value |
| Urban Level (LULC) | Percentage of Total Area |
| Transit Stations | Count within 0.5 Miles |
| AFV Fueling Stations | Distance to Nearest |

For transit stations, the count of stations present within 0.5 miles of the Block Group or Tract was used. The variables for contaminated sites and AFV fueling stations were calculated as the distance from the edge of the Block Group or Tract to the nearest point occurrence. The purpose of utilizing a distance measure over count in this case was to capture both areas containing, and in proximity to, the sites. For discrete polygon datasets such as flood zones, proportional area was utilized, calculated as the percentage that the area occupies of the Block Group or Tract. As for continuous data such as $PM_{2.5}$, blue space, and green space, the mean value within the Block Group or Tract was used.

As $PM_{2.5}$ levels were measured at individual EPA air quality monitoring stations, interpolation was required to estimate measures beyond the points of collection. To accomplish this, the mean annual concentration was calculated for 2019 and interpolated in Esri's ArcGIS Pro (Version 3.0.2) from 47 individual monitoring stations across New Jersey, New York, and Pennsylvania. Exploratory interpolation was conducted before the Empirical Bayesian Krigging tool was selected to most appropriately interpolate pollution levels beyond the points of measure. This method was chosen following a cross-validation investigation, wherein Empirical Bayesian Krigging produced the highest prediction accuracy, as conveyed by the root mean square error.

We also introduced level of urbanization as a control variable. Land use/land cover (LULC) imagery was collected from the United States Geological Survey (USGS) National Land Cover Database for 2019 as a proxy for this measure [69]. For the purpose of this study, we assign urbanization values by proportional area, dividing the area of 'developed' land (LULC values 21 to 24) by the total Block Group or Tract area. While LULC is not a perfect measure for urbanization, it nonetheless provides valuable insight into urban development efforts and is a popularly used proxy in the realm of environmental justice research [50].

The data for transit stations, AFV fueling stations, and contaminated sites were collected as shapefiles from the New Jersey Geographic Information Network (NJGIN) [70]. Transit stations included passenger train, bus, and light rail stations. AFV fueling station data included publicly available stations for biodiesel, compressed natural gas, ethanol, electric, and propane vehicles. As the contamination dataset included remediated sites, the data were restricted to those with the highest environmental stressor score. Air quality and flood

risk data were drawn from the Federal EPA and Federal Emergency Management Agency, respectively. Flood zones with the category AE were selected for this study, representing a 1% annual chance of flooding. Maps showing the spatial distribution of all environmental factors aggregated to the Census Tract level can be viewed in the Supplementary Materials (Figures S1–S8).

### 2.2.4. Social Vulnerability

Social vulnerability variables were established using data taken from the US Census Bureau's American Community Survey (ACS) at the Block Group level for 2019. To maintain consistent data sources between resolutions, all Tract-level Census data were derived from aggregated Block Group data with the exception of the Median Household Income, which was drawn directly from the Census website for the Tract level. Qualities associated with social vulnerability were chosen based on previous studies which identified characteristics that, as a result of logistical limitations or institutional marginalization, may hinder an individual's ability to respond to undesirable environmental circumstances, particularly in the context of the US [14,16,71].

In some cases, these limitations introduced by identity are more nuanced. For example, in the United States, the ramifications of institutions historically designed to discriminate against people of color were, in theory, eliminated following the Civil Rights movement. However, discriminatory policies such as redlining that were intended to keep people of color out of certain neighborhoods continue to impact the distribution of households in relation to hazards and resources, with hardships compounding in the face of emergencies [4,72,73]. In an attempt to capture these types of social vulnerability, a mix of characteristics were drawn from the ACS and used in this study. Excluding median household income, all social variable values were calculated as proportions of the category total. The finalized selection of social vulnerability variables is listed in Table 4.

**Table 4.** Social vulnerability measures—all values are proportion of category total except median income.

| Variable | Population from which Data was Drawn |
|---|---|
| Black or African American | General Population |
| Hispanic or Latino | General Population |
| With Individual(s) 65 and Over | Households |
| Education Below High School Graduate | Population 25 Years and Older |
| Median Income | Households |
| With a Disability | Population 20 to 64 Years Old |

### 2.3. Methodology

### 2.3.1. EJ Awareness

To address our research questions, we use social media data as the dependent variable, defining environmental justice awareness as Tweet (with EJ words) frequency in a Block Group or Tract, and investigate its relationship with environmental factors and socially vulnerable populations. While relying solely on geographic relationships fails to capture all nuances of environmental justice (particularly recognition and representation justice), our intention is to establish a foundation for analyzing these complexities. Our study area consists of 3029 Block Groups and 922 Tracts with population data across eight counties, and we expect that areas falling within the same municipal boundaries will have similarities in terms of their environmental justice levels. After aggregating the number of Tweets containing at least one environmental justice term, we find that 1457 Block Groups and 789 Tracts contain at least one Tweet.

### 2.3.2. Inverse Normal Transformation

Due to the high number of Block Groups and Tracts containing zero Tweets with environmental justice terms, a data transformation is warranted to normalize our depen-

dent variable. To proceed with appropriate models, we borrow ideas from the Inverse Normal Transformation (INT) strategy proposed in [44] that is commonly used in genome studies. This approach is able to transform the zero-inflated and highly skewed data to be approximately normally distributed. Detailed steps follow.

The zero-inflated and highly skewed Tweet count will be standardized by Block Group or Tract area and put through the rank-based INT strategy. Specifically, let $rank(c_i)$ denote the sample rank of $c_i$ when the measurements are placed in ascending order. The rank-based INT is defined as:

$$INT(c_i) = \varphi^{-1}\left[\frac{rank(c_i) - k}{n - 2k + 1}\right]$$

Here $\varphi^{-1}$ is the normal density function, $k \in (0, \frac{1}{2})$ is an adjustable offset, and $n$ is the sample size. By default, the Blom offset of $k = 3/8$ is adopted [44]. This transformation effectively removes the zero-inflation and makes the $INT(c_i)$ an approximately normally distributed variable. We refer to these values as "environmental justice Tweets" going forward.

### 2.3.3. Regression and Spatial Regression Models

The OLS regression was used to examine the relationship between our independent variables and environmental justice awareness, as it is a flexible linear regression model commonly used in environmental justice studies and can be easily compared to permutated models [14,20,22,74,75]. The OLS procedure relies on a linear model consisting of a constant, coefficients, and an error term derived from the selected independent and dependent variables. The coefficients are estimated by finding the model that minimizes the sum of squares between the predicted and observed values, in turn providing insight into the relationship between our observations. For our purposes, the OLS will be expressed as:

$$Y_i = \beta_0 + \beta_{i1} + \cdots + \beta_{ij} + \varepsilon_i$$

where $Y_i$ is INT transformed environmental justice Tweets in study area $i$, $\beta_0$ is a constant, $\varepsilon$ is the estimated error term, and each value of $\beta_j$ is a social vulnerability or environmental term. We are interested in exploring the relationship discovered between environmental and social factors and their associated environmental justice Tweet count. As such, the OLS model will provide a foundation for this examination.

However, to ensure our assumption of independence is met, the spatial nature of our data requires us to test for spatial autocorrelation. We can do so using Moran's I, a statistical measure frequently used in spatial analyses to assess the positive and negative clustering in a dataset [23,76,77]. For models where the spatial autocorrelation of the residuals is detected and is determined to be significant, a spatial statistical technique can be utilized to control such autocorrelation. We selected a spatial autoregression (SAR) procedure for the purpose of our study, as it allows us to identify whether the dependency detected stems from our dependent variables, error terms, or a combination thereof. In this way, we can identify the source of the autocorrelation, allowing us to further explore the relationship between social vulnerability, environmental factors, and our environmental justice Tweet measure. If spatial autocorrelation is detected, we use the Lagrange Multipliers (LM) test as our maximum likelihood-based statistic to determine whether a spatial lag or error specification will most effectively capture spatial autocorrelation in the residuals. As an example, the spatial error model (SEM) is as follows:

$$Y_i = \beta_0 + \beta_{i1} + \cdots \beta_{ij} + \varepsilon_i \varepsilon_i = \lambda \sum_j W_{ij}\varepsilon_i + \xi_i$$

where $\lambda$ is the coefficient of the autocorrelated term, $W_{ij}$ is the spatial weight matrix outlining the geographic relationship between neighboring Block Groups or Tracts, and $\xi$ is a vector of uncorrelated error terms.

## 3. Results

### 3.1. Environmental Justice Awareness

We first present the results of our broader analysis, investigating the relationship between socially sensed environmental justice awareness, remotely sensed imagery, and environmental justice factors through an OLS and SAR model.

We sought to conduct our analysis at two resolutions, Tract and Block Group. After examining the linear relationship between independent variables and the INT transformed environmental justice Tweet counts, the Block Group level was deemed unfit for analysis. Despite the transformation, the high number of Block Groups with zero environmental justice Tweets highly skewed the model, resulting in insignificant linear relationships. At the Tract level, all assumptions were met after introducing quadratic terms for several variables to achieve linearity. As such, the remainder of our analysis focuses on the Tract level. Additionally, the indexes for green and blue space selected for our model were the NDMI and MNDWI, respectively. R-squared and AIC values suggested the NDMI performed better than the NDVI by a very small margin, while MNDWI was clearly the best fit for our model.

After reviewing the distribution of Tweets with environmental justice terms, it also becomes apparent that our reliance on point data to estimate the geographic source of Tweets has resulted in the MAUP. This bias is the result of the nature of Census boundaries, where study areas like Tract and Block Groups are drawn to create a relatively equal distribution of population estimates, resulting in areas of high population density displaying smaller Census polygons. This is clearly observed in our study area, where high-density urban areas in the east have smaller Block Groups and Tract, while low-density areas in the west have larger Block Groups and Tracts. As a result, although the highest number of Tweets are present in the eastern part of our study area, this density is lost when joining to Census polygons due to the relatively small size of the Tracts and Block Groups. Instead, it appears as if the distribution of Tweets is rather scattered, with large Tracts and Block Groups containing higher values. We intend for our area weighting of the dependent variables prior to the INT transformation to help control this bias.

With these details in mind, we turn to our Tract level OLS model's results (Table 5). First, we note that our control variable, urban levels, demonstrates a *p*-value above 0.05. Examining the social vulnerability variables, the results show all coefficient estimates are negative. The variables for Black or African American, Hispanic or Latino, and households with an individual over 65-years-old are particularly of interest, with *p*-values below our 0.05 alpha. The *p*-value for education below high school graduate approaches this alpha level, but remains slightly above it. Looking next to the relationships between remote sensing variables, we see only the green space (NDMI) variable demonstrates a statistically significant *p*-value with our social sensing dependent variable. Notably, this is also one of only two environmental variables demonstrating a significant *p*-value and a negative coefficient, suggesting that as green spaces decrease, environmental justice Tweets increase. On the other hand, the *p*-value for blue space (MNDWI) variable does not fall below the alpha. This may indicate a lack of concern or habitual ignorance, meaning residents do not often associate water bodies with environmental benefits. Finally, examining the additional environmental variables, AFV stations was the only measure with a *p*-value below the 0.05 alpha, with transit stations only nearly falling above this threshold. Contaminated sites proximity, $PM_{2.5}$, and flood zone area are all not statistically significant, suggesting the public awareness of exposure to potential contaminants or association with flood risk and environmental injustice do not present strong ties in our investigation. Similar to blue space, we attribute this as a possible result of habitual ignorance, suggesting residents often are not aware of the proximity to contaminated sites, flood risk, or air pollution.

**Table 5.** OLS results—Tract (multiple R-squared = 0.2479).

| Coefficient | Estimate | Std. Error | t-Value | *p*-Value |
|---|---|---|---|---|
| Intercept | 2.3500 | 1.28200 | 1.833 | 0.06716. |
| Black or African American | −0.5229 | 0.18260 | −2.864 | 0.00428 ** |
| Hispanic or Latino | −0.5267 | 0.26660 | −1.976 | 0.04851 * |
| With Individual(s) 65 and Over | −4.2780 | 1.39800 | −3.061 | 0.00227 ** |
| Quadratic—With Individual(s) 65 and Over | 5.3940 | 2.23300 | 2.416 | 0.01589 * |
| Education Below High School Graduate | −1.0250 | 0.59560 | −1.720 | 0.08577 † |
| With a Disability | −2.4510 | 1.56800 | −1.563 | 0.11837 |
| Quadratic—With a Disability | 9.6370 | 5.12700 | 1.880 | 0.06048 † |
| Median Income | $-1.43 \times 10^{-6}$ | $1.175 \times 10^{-6}$ | −1.218 | 0.22349 |
| Contaminated Sites | $1.89 \times 10^{-5}$ | $1.535 \times 10^{-5}$ | 1.230 | 0.219 |
| Flood Zones | −0.1567 | 0.19920 | −0.786 | 0.43193 |
| PM$_{2.5}$ | −0.1020 | 0.17120 | −0.596 | 0.55153 |
| Green Space (NDMI) | −9.9120 | 2.07700 | −4.772 | 0.00000213 *** |
| Transit Stations | 0.0020 | 0.00107 | 1.892 | 0.05887 † |
| Urban Level (LULC) | 0.2848 | 0.17300 | 1.647 | 0.10001 |
| Blue Space (MNDWI) | −1.6010 | 1.76700 | −0.906 | 0.36516 |
| AFV Fueling Stations | $-9.93 \times 10^{-5}$ | $2.020 \times 10^{-5}$ | −4.915 | 0.00000105 *** |

Note: * Significance level: † means *p*-value is less than 0.1, * means *p*-value is less than 0.05, ** means *p*-value is less than 0.01, and *** means *p*-value is less than 0.001.

The spatial effects in these models provide further insights. We test our dependent variable for spatial autocorrelation in R using Moran's I. We constructed a row standardized neighborhood list using the *nb2listw* function from the 'spdep' package. We calculated a Moran's I value of 0.2510 with a *p*-value near zero, suggesting significant positive global spatial autocorrelation is present in the Tweet distribution. Examining the local Moran plot for our model, we further observe a positive trend in local spatial autocorrelation, suggesting clusters of high values surrounded by high values. As discussed, when spatial autocorrelation is detected, the regression assumption of data independence is violated and should be addressed to draw more appropriate conclusions. We used the Lagrange Multipliers (LM) test to select between a spatial lag or error specification as commonly practiced [78]. The LM results point to a spatial error model (SEM) as the most appropriate for our analysis. This suggests the spatial dependency present in our model stems primarily from the model's error term, pointing to spatial autocorrelation in factors not included in our analysis.

The SEM model results (Table 6) show this approach appears to improve upon the standard OLS. The AIC for the OLS is 2312.2 compared to 2293.4 for the SEM. As the SEM AIC value is lower, we conclude that controlling for spatial autocorrelation has improved the estimation of the relationship between our independent and dependent variables. Looking more closely, the SEM model shows similar results to the OLS in several aspects, with some notable differences. For all variables, the positive or negative nature of the coefficient estimates for statistically significant variables did not change, suggesting that both models captured the essence of the EJ awareness and social and environmental factors. Examining the social variables, the magnitude of nearly every variable increased or shifted only slightly. The variable for households with an individual over 65 was the only notable exception, with its coefficient moving closer to zero to a greater extent than any other variable. Along this same line, the *p*-value of nearly all variables decreased or stayed approximately the same. Median household income in particular now has a *p*-value below the alpha. This again excludes the variable for households with an individual over 65, which had its *p*-value increase by over ten-fold, while still maintaining marginal significance at a level just below the alpha. This suggests spatial dependency played a non-negligible role in the relationship between this variable and environmental justice Tweets. Controlling this phenomenon has reduced the impact of age in our model.

**Table 6.** Spatial error model results—Tract (AIC: 2293.4, AIC for OLS: 2312.2).

| Coefficient | Estimate | Std. Error | t-Value | *p*-Value |
|---|---|---|---|---|
| Intercept | 2.1371 | 1.60720 | 1.3297 | 0.1836151 |
| Black or African American | −0.6967 | 0.20554 | −3.3896 | 0.0006999 *** |
| Hispanic or Latino | −0.7503 | 0.28379 | −2.6438 | 0.0081972 ** |
| With Individual(s) 65 and Over | −2.7615 | 1.40550 | −1.9647 | 0.0494483 * |
| Quadratic—With Individual(s) 65 and Over | 3.3028 | 2.21300 | 1.4924 | 0.1355912 |
| Education Below High School Graduate | −0.9161 | 0.61025 | −1.5011 | 0.1333302 |
| With a Disability | −2.6050 | 1.56700 | −1.6624 | 0.0964372. |
| Quadratic—With a Disability | 9.7751 | 4.98610 | 1.9605 | 0.0499424 * |
| Median Income | $-2.72 \times 10^{-6}$ | $1.2129 \times 10^{-6}$ | −2.2391 | 0.0251519 * |
| Contaminated Sites | $2.00 \times 10^{-5}$ | $1.8002 \times 10^{-5}$ | 1.1127 | 0.2658403 |
| Flood Zones | −0.1600 | 0.21251 | −0.7527 | 0.4516273 |
| PM$_{2.5}$ | −0.0731 | 0.21568 | −0.3391 | 0.7345236 |
| Green Space (NDMI) | −11.3270 | 2.20140 | −5.1456 | 0.0000002667 *** |
| Transit Stations | 0.0027 | 0.00118 | 2.2763 | 0.0228305 * |
| Urban Level (LULC) | 0.1850 | 0.19290 | 0.9589 | 0.3376311 |
| Blue Space (MNDWI) | −2.3341 | 1.89260 | −1.2333 | 0.217457 |
| AFV Fueling Stations | $-9.55 \times 10^{-5}$ | $2.2798 \times 10^{-5}$ | −4.1876 | 0.00002819 *** |

Note: * Significance level: * means *p*-value is less than 0.05, ** means *p*-value is less than 0.01, and *** means *p*-value is less than 0.001.

Turning to the remote earth observation and additional environmental variables, the observable changes to our model are relatively modest. For most variables, *p*-values changed only slightly. Notable exceptions in this case include the control variable urban area, which increased significantly, and transit stations, which dropped below the alpha. Along this same line, coefficient estimates remained virtually the same in most cases, with only slight variation between models. Overall, our results suggest the SEM has refined our model and provided some clarification regarding the relationships between environmental justice Tweets and our independent variables. While upholding most of the initial statistical relationship revealed by the OLS model, the spatial model provides a more robust calibration that is better suited for our data.

### 3.2. Impact of Smoke on Tweets

Next, we present results from our pointed investigation, focusing on scrutinizing socially sensed data produced throughout a natural disaster event.

Our analysis of this data during the Spring Hill Wildfire was intended to show how discussions on Twitter shifted spatially and thematically during a period of poor air quality stemming from a natural disaster. Utilizing the short-wave infrared and near-infrared bands of the MODIS imagery, we are able to clearly identify the path of smoke travelling north from the wildfire, passing primarily through the center and east of our study area. The extent and impact of the smoke is confirmed by the PM$_{2.5}$ data interpolated for the same day, with the areas falling within and directly adjacent to the smoke boundary demonstrating the highest concentrations. News reports further corroborate these data, with some indicating that smoke could be seen and smelled as far north as Bergen County (Figure 1), located in the northeast corner of our study area [79–81]. One report stated that air quality in Newark (Essex County, the eastern part of our study area) maintained unsafe levels over a 24 h period beginning in the evening on 30 March [79].

A total of 53,006 Tweets were drawn from 30 March to 1 April 2019, but only 2398 had geographic coordinates and fell within our study area. Of these Tweets, 629 appeared within the smoke boundary digitized using the MODIS imagery. However, reviewing these Tweets showed little to no discussion on smoke, air quality, or the fire itself. In fact, in the whole study area, the word *fire* appeared 12 times, *air* appeared 3 times, and *smoke* appeared 3 times, but the content of these Tweets showed that none of them related to air quality or the Spring Hill Wildfire in any way. When including Tweets surrounding the

study area, a total of 15,807 Tweets with geographic coordinates appear, but a similar trend emerges, with *fire* appearing 49 times, *air* appearing 28 times, and *smoke* appearing 27 times. Again, a manual review shows that no Tweets appear to reference the wildfire or air quality specifically, with users instead sharing sentiments like "Spring is in the air". We finally returned to the original set 53,006 Tweets, many of which appeared outside of the study area or had no coordinate data. In this expanded dataset we found the word *fire* 172 times, *air* appeared 170 times, and *smoke* appeared 93 times, but manual investigation revealed only three Tweets relating to the fire or its smoke. These Tweets each mentioned the smoke disrupting visibility specifically, and one mentioned the Spring Hill Fire by name. No other Tweet's content could be attributed to the subject.

Aligning with our broader analysis results, this further suggests that the general public's awareness of air quality, especially as a result of a short-term event like the Spring Hill Wildfire, often does not stir many social sensing responses, at least on the Twitter platform. Short-term events, while having significant impact on environmental quality, especially for vulnerable communities, might not align well with the quick "mentioning" nature of social sensing platforms. Still, considering social sensing platforms' wide reach and coverage, policymakers or environmental agencies might benefit from taking advantage of these data sources to purposefully advocate and disseminate information through the platforms, particularly during environmental events. This sort of activity seems to be lacking at present.

## 4. Discussion

To review, we intend for this study to answer the following research questions: (1) Is there a discernible relationship between environmental justice factors inferred from remotely sensed earth observations, traditional governmental sources, and social sensors? (2) How can we best model this relationship using Twitter data, social vulnerability measures, and environmental factors? (3) What are the potential consequences of leaning on these remote sensing datasets to draw conclusions about environmental justice? We will explore each of these questions in the discussion below.

### 4.1. Relationships between Environmental Justice, Remotely Sensed Imagery, and Social Sensing

First, we review our broader analysis, examining the relationship between remotely sensed earth observations, additional environmental factors, socially vulnerable populations, and socially sensed environmental justice activity. Given that the SEM specification provides a better fit for our data, we will focus our discussion on its results.

Beginning with the relationships between remote sensing datasets, we found that the MNDWI did not appear statistically significant, but the NDMI measure exhibits a significant negative relationship with the environmental justice awareness proxy of Tweet counts. While our investigation of spectral indexes showed that all vegetation measures demonstrated statistical significance in our model, NDMI was the best fit, narrowly surpassing even the popularly used NDVI. We contend that this ubiquitous statistical significance among vegetation indexes helps to answer our first research question by demonstrating a measurable relationship between social media big data and vegetation derived from remote sensing imagery. We have further found that, in the case of our own study, the NDMI acts as the measure of best fit for this modeled relationship, addressing research question two. However, there is room for improvement in this model, as the relationship between factors is complex and imperfectly reflects environmental justice realities. We will unpack this idea below in order to answer our third research question.

It is perhaps fitting that vegetation measures have a complex relationship with environmental justice. On the one hand, greenery is a popular topic culturally, politically, and socially. Broadly speaking, it plays an important role in environmental justice due to the insinuated positive mental and physical health impacts [82], services provided by public open space [83], and recorded inequitable distribution [23,84,85], each of which were exacerbated during the global COVID-19 pandemic. With this in mind, we would expect

ecosystem services at large to be a topic of popular discussion, justifying a higher volume of environmental justice Tweets. On the other hand, ecosystem services are not perfectly modeled by spectral indexes. For example, the popularly used NDVI [58,86–89] is not always the most appropriate measure (as we have shown is this case in our own model) and is similarly, not a perfect proxy for evaluating ecosystem services.

Previous researchers have noted that indexes like the NDVI can be effective at communicating the presence and intensity of vegetation, but do not inherently provide information about the type of vegetation detected, the vegetation's health, or the ecosystem service being provided as a result [3,57,86]. To this end, the details of the relationship between environmental justice and an NDVI cannot be reasonably assumed without a deeper investigation. In a recent study by Schwarz, Berland and Herrmann [86], greenery captured over the years by an NDVI in Toledo, Ohio was often associated with overgrown vegetation and unkempt lawns emerging at abandoned sites. In this case, socially disadvantaged populations were positively correlated with increases in housing vacancy rates and NDVI values. This meant that vegetation was higher in marginalized communities, but further investigation showed this vegetation was in the form of overgrown yards and unwanted weeds, neither of which would provide the ecosystem services often associated with exposure to greenery.

We are not implying that this relationship is the case for our own study, but rather that the role spectral indexes play on the presence of environmental justice or injustice is not perfectly modeled when relying solely on remotely sensed imagery. In fact, we contend that our results demonstrate the potential for pairing traditional remotely sensed earth observation datasets with social sensing information. Our study suggests that there is an opportunity to mitigate inaccuracies and biases emerging from spectral indexes like the NDMI and MNDWI by pairing measures with observations made by individuals on the ground. Whereas the extent of ecosystem services provided by vegetation cannot be extrapolated from our NDMI, it can be paired with social sensing sources like social media to identify areas where environmental justice discussions overlap. In our particular study area, this manifests in the negative relationship observed between greenery and environmental justice awareness, suggesting a very indicative phenomenon in highly urbanized areas. Less access to or presence of surrounding green spaces prompts people to discuss to a greater extent, suggesting a potential pattern of environmental injustice when examining the NDMI distribution in urbanized neighborhoods.

This synergistic relationship between social sensing and remotely sensed imagery is further reflected in our pointed investigation of smoke and wildfire impact on Twitter activity. The use of MODIS imagery to identify the path of smoke effectively outlined areas experiencing the densest coverage, as corroborated by $PM_{2.5}$ data interpolated for the same day. However, no Tweets anywhere within the study area appear to mention the Spring Hill Wildfire or the impacts on local air quality. Perhaps more surprisingly, this trend persists even when expanding Tweet criteria to include neighboring areas and Tweets without geographic data. News reports from the first and second day of the fire expressed that the smell of smoke was present even in New York City [90], but our investigation fails to capture much of this sentiment, even when including Tweets in those areas. We consider two theories that may explain this lack of Twitter data.

First, this lack of discussion on environmental conditions in the face of smoke, wildfire, and reduced air quality aligns with the results from our broader analysis, which showed that flood risk, contaminated sites, and particulate matter were not statistically significant variables in our model. In other words, our analysis of environmental justice awareness showed that the risk of natural disaster and exposure to contamination failed to generate a social sensing response on Twitter. It is perhaps understandable, then, that a wildfire event that resulted in exposure to contamination in the form of smoke failed to trigger a social sensing response. This idea is further echoed by the findings of a study by Xu et al. on perceptions of air quality in Beijing [91]. The authors interviewed individuals who had lived in the community for at least two years and found that 41 out of 43 residents

recognized the air was polluted. However, 35 interviewees also expressed that they felt slight or no concern regarding the negative impacts of their surrounding air quality. The reasons provided for those feeling a lack of concern included reports that individuals felt powerless to make a meaningful change, that they were not experiencing immediate health impacts, and that there were simply more pressing concerns such as food security and housing costs, alongside several other explanations. In this case, community members demonstrated a knowledge of an existing environmental hazard, but generally did not deem the risk of concern.

It is also possible that the short-lived, dynamic nature of the event may have reduced social sensing responses. As factors like wind, rain, and fuel change, the dispersion and impact of smoke would follow suit, potentially getting better and worse each hour. Our analysis examined all three days of the Spring Hill Fire, but assuming smoke conditions were changing along with these characteristics, it is possible the negative air quality did not persist for long enough to warrant a response from Twitter users. Alternatively, sentiment expressed over Twitter regarding environmental qualities during the wildfire may in fact exist, but is simply not captured due to technical and methodological constraints. For example, the nature of the Twitter API is such that only Tweets from public accounts are accessible in order to maintain the privacy of users. As a result, an unknown number of Tweets from individuals with private accounts are omitted from our analysis. Similarly, the medium with which users discuss topics like air quality may not be explicitly conveyed through text, but rather a combination of images and emojis. In this way, a Tweet may acknowledge air quality or wildfire in a manner that would not be detected in our textual analysis despite its relevance. More complex expressions such as sarcasm and metaphors are also harder to detect and may result in missed Tweets.

Given that less than 0.01% of the 53,006 public Tweets textually referenced the environmental impacts of the wildfire, it is perhaps fair to say that it is unlikely these technical shortcomings would drastically change our analysis. In any case, our results clearly point to a disconnect between Twitter data and smoke exposure. These insights are important in the context of exposure to poor air quality and disaster risk, but nonetheless demonstrate a connection between remote sensing imagery and social media data.

### 4.2. Big Data, Additional Environmental Factors, and Social Vulnerability

We turn next to our additional environmental factors, allowing us to better contextualize the relationship between socially sensed environmental justice and traditional measures.

First, it appears there is a positive relationship between the urbanization control variable (urban land use land cover percentage) and Tweets of environmental justice, but the measure does not pose a statistically significant relationship with environmental awareness in either model. Considering the highly urbanized landscape in northern New Jersey, this result might not be as unexpected. Since urbanization levels in the study area are quite high, the variation in urbanization level, as expressed as the land use land cover percentage of impervious surfaces, might not be sufficient to provide statistical power to explain the variation of the Tweet counts relating to environmental justice.

The other environmental factors including $PM_{2.5}$, flood zone, and proximity to contaminated sites all do not show statistically significant relationships with the environmental justice Tweet counts. We argue that this is likely because of the less tangible nature of these environmental factors. Indeed, compared to greenery, which is immediately tangible, having a sense of particulate matter in the air, percentage of flood zone, access to waterbodies, or proximity to contaminated sites is much less sensible. Twitter, as one of the many social sensing platforms, will have a lower sensitivity to capturing these factors in daily Tweets.

Regarding the negative relationship between AFV fueling stations and environmental justice Tweets, our results suggest that as AFV fueling stations grow nearer, the number of Tweets containing environmental justice terms goes up. In other words, a greater presence of investment in sustainability, captured as AFV fueling stations, is associated with a greater awareness of environmental justice levels. We argue that this relationship may be the result

of an awareness of environmental factors generally in these communities. Whether this investment is the result of government, business, or community advocacy, it is fair to say that the presence of AFV fueling stations stems from support at some level. As a result, this interest and subsequent presence of AFVs on community roads likely fosters a sense of environmental consciousness [92]. It is reasonable to assume that this consciousness would be reflected on social media, captured in our analysis as individuals discussing environmental justice topics.

Additionally, the explanatory variable for transit stations is a particularly nuanced measure, having a juxtaposed impact on communities. Community members near transit stations benefit from the additional mobility and reductions in air pollutants in the long-term [93], but risk disproportionate exposure to harmful pollutants in the short-term. Moreover, the concentration of public transit stations might also point to a relatively dense and more socioeconomically vulnerable community [94–96]. Our model suggests that despite the complex relationship between transit stations and our community, higher concentration of public transit stations certainly poses a tangible and sensible signal that heightened residents' environmental justice awareness. This is demonstrated by the variable for transit station's significant relationship with the count of environmental justice awareness Tweets.

At this point, we have established that there is an apparent relationship between obviously sensible environmental factors (greenery, AFV fueling stations, and density of transit stations) and the discussion of environmental justice on Twitter. However, our model results do not suggest that there is awareness of environmental injustice among socially vulnerable populations. Instead, the modeled results show that (Table 6), as there is an increase in the proportion of population that is Black or African American, is Hispanic or Latino, lives in a household with someone over 65 years old, has a disability, or has below a high-school-level education, the discussion of environmental justice on Twitter decreases. Median household income is the only variable to suggest otherwise, with a negative coefficient value, meaning that as household income decreases, the number of Tweets using environmental justice terms increases.

Of the five social factors, the coefficients of Black or African American, Hispanic or Latino, households with an individual over 65, and median income show statistical significance (Table 6). Of particular note, we recognize from our visual exploration of each variable in ArcGIS Pro that proportions of the population that are Black or Hispanic are disproportionately highest in urban centers found in the east of our study area, in areas such as Newark, Paterson, and other cities in Hudson County. As discussed, Newark in particular is a well-known environmental injustice community [8]. As such, it would be expected that these known exposures to environmental injustices would correspond with an increase in environmental justice terms on Twitter. The negative coefficient estimates suggest otherwise.

We theorize there are two possible explanations for the negative relationships observed between the social factors and environmental justice Tweets. On one hand, these relationships may mean that there is a genuine lack of awareness or acknowledgement among these communities that environmental injustice is occurring. This may be the result of extended exposure to hazards and limited access to resources leading to a normalization of the experiences. In other words, individuals that are exposed to hazards or kept from resources adjust to the point of no longer recognizing or acknowledging their own plight, hence forming a habitual ignorance of their environmental injustice. In the case of communities like Newark that have been experiencing environmental injustice for decades, it is reasonable to assume individuals who grew up in these places would view their experiences as normal. If nothing else, community members would likely stop discussing negative things in their community on Twitter daily after experiencing them for their entire life. In either scenario, we would expect to see a lack of environmental justice Tweets.

As mentioned above, the result of Xu et al.'s study on public perception of air quality offers a similar explanation, as residents demonstrated a lack of concern for the subject

as a whole despite knowledge of the risks [91]. Interviewees expressed a sort of apathy, considering air quality a lower concern in most cases. It is possible we are witnessing a similar phenomenon in our own study, wherein individuals living in environmental justice communities like Newark and Patterson recognize the existence of injustice, but do not express concern. Whether they feel a sense of powerlessness, are prioritizing other concerns, or simply believe the factors are not worthy of attention, if residents feel the injustices they suffer are not a major concern, they would likely not Tweet about them.

On the other hand, the negative relationship we have observed in our model may also be a symptom of the MAUP. We see this phenomenon occur particularly in the highly urbanized zones in the east of our study area, which as mentioned also contain the highest proportions of people of color in our study. While we attempted to mitigate the impact of the MAUP by weighting counts by area, this did not correct for Tracts and Block Groups that were assigned zero Tweets despite the high volume of points in these urban areas. This bias may have resulted in an undercounting of Tracts and Block Groups with socially vulnerable populations, leading to a negative relationship between these variables and environmental justice Tweet count. With either explanation of these relationships, the result is the same— the voices of socially vulnerable populations are underrepresented in environmental justice discourse occurring on Twitter. This result merits further investigation to facilitate a more in-depth analysis and understanding of communities that are particularly vulnerable.

## 5. Conclusions

In an era characterized by unprecedented technological innovation and a growing urgency to address environmental inequities, the convergence of big data analytics and remote sensing offers an unprecedented opportunity to unravel complexities posed by environmental injustice faster than ever before. At this time, environmental justice is a profoundly important concern that is being increasingly placed at the top of policymakers' agendas, but initiatives and policies are reactive in nature. The result is often a case of 'too little, too late,' leaving marginalized populations displaced or living in unsafe conditions for extended periods of time. Community-based science offers an alternative, allowing those experiencing the injustices in their daily lives to catalog and report the data themselves [24,97–99], but these efforts are narrow in scope and unfairly place the burden of proof on residents. Big data sources such as social sensing information derived from Twitter and remote sensing imagery may yet serve a dual purpose in this context, acting as an early warning system for injustices and a corroborating source for community scientists seeking to raise the alarm.

Within this context, this study ventures into unexplored terrain, where the realms of remote sensing imagery and social media data intersect, seeking to untangle the intricate relationships between environmental hazards, ecosystem services, and social vulnerability. Our analysis indicates that there is a negative relationship between the number of Tweets utilizing environmental justice terminology and the presence of ecosystem services in the form of green spaces (most effectively captured by the NDMI), suggesting a synergy between the datasets and a broad awareness of injustice. However, there is simultaneously a negative relationship between socially vulnerable populations and Tweets with environmental justice words. This suggests that, generally, there is discussion on Twitter about injustice when resources are not present, but the voices of vulnerable populations are often unaccounted for, very likely as a result of urban bias and a habitual lack of awareness or concern for injustices. This latter theory is echoed in the results of our case study, demonstrating a lack of discussion on Twitter during the state's largest wildfire of the year. Overall, our findings suggest that a meaningful relationship is present between social sensing and remote earth observation data in the context of environmental justice, but capitalizing on this intersection may perpetuate inequities if precautionary measures are not taken.

Our research is not without its limitations. First, data and computational constraints limited the number of Tweets which could be drawn for this study and required data

aggregation. With superior hardware, additional data, and sufficient knowledge of machine learning techniques, there may have been an opportunity to parse all words across a longer timeframe to fit a more precise model which could operate beyond the limitations of aggregation. Second, we recognize the niche, dynamic social media landscape presented by Twitter that likely introduces inequities into our model. We utilize Tweets containing environmental justice terms as a proxy for community discussion, assuming that this platform hosts honest conversations on lived experiences by a wide variety of community members. In reality, social media in general is likely to generate a wide variety of discussions, likely by young, technology-literate individuals with downtime and access to a smartphone or computer. In fact, some third-party reports have shown that around 62% of users are below the age of 34 [100] and over 60% of Twitter users identify as male [101].

Along this same line, the accuracy of coordinates collected for Tweets is relatively unclear. Twitter's API website explains that coordinates come from GPS-enabled devices and should represent exact location, but can be assigned in some circumstances [102]. Content produced on Twitter also varies significantly from user to user, and these habits are impacted by trending topics, world events, and even ownership of the platform itself, each of which are often unexpected and unpredictable. Our model attempts to capture a snapshot of Twitter activity and make assumptions based on the observed trends in that time period, but future trends are likely to differ in frequency, topic, and impact. Our exploration nonetheless suggests that there is great potential. For this reason, future researchers might consider utilizing other socially sensed big data sources as an alternative or supplement to social media. Sources such as internet search engines may be particularly well suited for this purpose, with tools like Google Trends being used to capture different facets of spatial human dynamics through online search query frequency [103–105]. Although social engines offer opportunities to capture information generated by a more diverse userbase regarding their intentional information gathering efforts, as with all data sources, they present their own challenges, specifically in terms of the ambiguity related with the broad concept of environmental justice. For this reason, our current study did not incorporate the search engines data into our study, though an immediate next step will certainly attempt to validate the potential of search engine data in future environmental justice studies, as the awareness of environmental justice is rapidly growing.

Third, exploring environmental justice through remote sensing imagery analysis and Tweets is an endeavor that will vary geographically and temporally. Although it is a subject that is relevant across the globe, environmental justice is highly contextually dependent [2], and as such, discussion (online and offline) on the subject will vary by location and time period. While New Jersey's diversity and legal recognition of environmental justice might make for an ideal case study, the extent to which this subject is broached in other states likely varies, requiring a more careful examination.

Despite these limitations, we believe our research takes a vital step towards critically investigating environmental injustice in the era of remote sensing and big data. We expect the extensive nature of these data, paired with emerging remote sensing data acquisition technologies (drone and hyperspectral images in particular) and advanced analytical methodologies (such as Bayesian spatiotemporal analytical framework [37]), may provide novel techniques for future research capturing information on these subjects. We simultaneously urge researchers to consider the equity implications stemming from big data in all its form, particularly regarding sources popular in research not analyzed here such as Google Trends [106]. Overall, this interdisciplinary exploration not only signifies a leap forward in remote sensing science, but also heralds a new era of comprehensive insights that extend beyond the boundaries of traditional research methodologies. Our hope is that these advancements might pave the way for a future in which environmental justice is a priority in more than just name.

**Supplementary Materials:** The following supporting information can be downloaded at: https://www.mdpi.com/article/10.3390/rs15235510/s1, Figure S1: Study Area Census Tracts—Urban Levels; Figure S2: Study Area Census Tracts—Transit Stations; Figure S3: Study Area Census

Tracts—Mean PM$_{2.5}$; Figure S4: Study Area Census Tracts—Greenery Index (NDMI); Figure S5: Study Area Census Tracts—Water Index (MNDWI); Figure S6: Study Area Census Tracts—Flood Zones; Figure S7: Study Area Census Tracts—Contaminated Sites; Figure S8: Study Area Census Tracts—AFV Fueling Stations.

**Author Contributions:** Conceptualization, C.K. and D.Y.; methodology, C.K. and D.Y. software, C.K.; validation, C.K. and D.Y.; formal analysis, C.K. and D.Y.; investigation, C.K.; resources, C.K. and D.Y.; data curation, C.K.; writing—original draft preparation, C.K.; writing—review and editing, D.Y.; visualization, C.K.; supervision, D.Y.; project administration, D.Y.; funding acquisition, D.Y. All authors have read and agreed to the published version of the manuscript.

**Funding:** This research was funded by the US Department of Housing and Urban Development, grant number NJLTS0027-22.

**Data Availability Statement:** The data presented in this study are available on request from the corresponding author. The data are not publicly available due to research grant restrictions.

**Conflicts of Interest:** The authors declare no conflict of interest.

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
