# Peer review of "Bridging the Gap: Analyzing the Relationship between Environmental Justice Awareness on Twitter and Socio-Environmental Factors Using Remote Sensing and Big Data"

_remotesensing, doi:10.3390/rs15235510_

Round 1
Reviewer 1 Report (New Reviewer)
Comments and Suggestions for Authors
This study examines the interrelationship between online discussions or tweets on environmental justice terms and local environmental characteristics derived from remote sensing data. It finds that although there is a negative relationship between green space distribution and the number of Tweets with environmental justice terms, the distribution of Tweets was not aligned with the distribution of vulnerable populations. The paper is well written and the results are informative with valuable insights on this particular issue. I only have a comment for minor revision.
The datasets utilized in this study are mostly spatial data, either from remote sensing images or social media. In addition, there has been discussion on some methodological limitations such as the MAUP problem. I would suggest the authors add a couple of maps to visualize the spatial layout of these environmental characteristics in these northern counties of New Jersey. (lines 190-191). In addition, whether Google Trends would be an alternative data source for environmental justice awareness would be worth mentioning.
Author Response
Response: Thank you for taking the time to conduct this review. We sincerely appreciate the time you have dedicated to this task and the kind words you have shared. Regarding your first comment, we agree that the inclusion of maps would help to better visualize characteristics discussed in this study. As such, we have added a general map of the study area to Section 2.1 as well as maps of each environmental factor to a Supplementary Material attachment. Regarding your second comment, we also recognize the potential opportunities of utilizing Google Trends data for environmental justice awareness. We have added mention of this approach to the Conclusion Section (Section 5) and are, in fact, working on a follow-up study using Google Trends data at this time as well!
Reviewer 2 Report (Previous Reviewer 2)
Comments and Suggestions for Authors
The paper addresses a complex problem – to explore environmental justice awareness and its factors. It first develops a measure of environmental awareness based on relevant social media posts, carefully screens social-environmental variables and develops measures of the variables, and finally investigates the relationships between awareness and all the variables using two different models. I am very impressed by the research design which is not only logically sound, but every step and every factor are carefully examined and thoroughly discussed. Although the final result does not completely confirm the initial hypothesis that a lack of ecosystem services and prevalence of socially vulnerable populations and environmental hazards will be positively related to the number of relevant tweet posts, factors including greenery, population composition, and urban level are found to be significantly related to environmental awareness. The paper also thoroughly discussed all variables and their impacts on environmental awareness. The research integrates a variety of data sources and technologies in solving the complex problem. I believe the paper will make a significant contribution to the subject area.
Author Response
Response: Thank you very much for your careful review and thoughtful comments!
Reviewer 3 Report (Previous Reviewer 3)
Comments and Suggestions for Authors
This is the second version of the research article with ID remotesensing-2531311. The authors carefully reviewed the article and did a good job following up on the comments and suggestions provided by the reviewers. The article clearly improved and I have no concerns recommending it for final publication.
Author Response
Response: Thank you for taking the time to conduct this review, we sincerely appreciate your efforts!
Reviewer 4 Report (New Reviewer)
Comments and Suggestions for Authors
Thank you for your work on analyzing the relationship between environmental justice awareness on Twitter and socio-environmental factors using remote sensing and big data with an aim to bridge the gap. This study, "Bridging the Gap: Analyzing the Relationship Between Environmental Justice Awareness on Twitter and Socio-Environmental Factors Using Remote Sensing and Big Data," already incorporates a variety of datasets, including social sensing data, remotely sensed imagery, additional environmental factors, and social vulnerability data. However, authors should consider incorporating several other important datasets into such a study to enhance its comprehensiveness and depth. I recommend including some additional datasets with their potential importance in this study and revising the manuscript's conclusions.
Comments on the Quality of English LanguageMinor edits would be required.
Author Response
Response: Thank you for agreeing to conduct this review and for your insightful comments. We recognize that the inclusion of additional datasets may help to further improve the comprehensiveness of the overall study. In seeking to address this note, we reviewed the impact of including the following additional factors: groundwater contamination, brownfields, parks, landfills, and alternative fuel vehicle (AFV) fueling stations. In this process, we collected the data and reviewed assumptions of linearity and covariation. Only the AFV station measure was deemed an appropriate addition to the final model following these tests, and as such, was added, described in greater depth, and discussed in the results section. We have added text describing the testing of these additional factors (Section 2.2.3), updated the results tables and descriptions (Section 3.1), and made additions to the discussion section (Section 4.2). We have also made minor edits throughout the paper in order to improve clarity in general. We thank you again for these comments, particularly as the inclusion of this additional variable has furthered insights gathered from our investigation.
Round 2
Reviewer 4 Report (New Reviewer)
Comments and Suggestions for Authors
Thank you for revising the manuscript.
Comments on the Quality of English LanguageMinor editing of the English language required.
This manuscript is a resubmission of an earlier submission. The following is a list of the peer review reports and author responses from that submission.
Round 1
Reviewer 1 Report
Comments and Suggestions for Authors
The authors have worked diligently in their revisions, and I respect their tenacity. They have responded to previous comments by adding ground level monitoring data. This has the unfortunate effect of further diluting the modest remote sensing content. I think this is a rare case of the paper getting worse as a result of responding to the reviewers comments. However, the fundamental problem at this stage is the diverse and complex analysis, only a small part of which relates to remote sensing. Moreover, there is no clear answer to the research questions. These ambitious and over-generalized research questions should be tightened. I suspect that there is good research paper (or two) that could be generated from the data. The paper needs to be reduced in length by about half. Once rewritten the authors may wish to consider submitting to a geography or planning journal. As applied research, the paper should be addressed to decision-makers, with recommended courses of action. If the authors wish to produce something specifically relating to remote sensing I suggest looking at different ways of determining green and blue space, other than NDVI. They acknowledge weaknesses of the NDVI approach and state that conventional approaches use NDVI, so are there better alternatives? They could also test their belief that a single scene or mosaic is sufficient.
My recommendation is that the paper be rejected, and that there be no further encouragement to resubmit.
Comments on the Quality of English Language
.
Author Response
Reviewer #1
The authors have worked diligently in their revisions, and I respect their tenacity. They have responded to previous comments by adding ground level monitoring data. This has the unfortunate effect of further diluting the modest remote sensing content. I think this is a rare case of the paper getting worse as a result of responding to the reviewers comments.
Response: Thank you for taking the time to conduct this review. We appreciate your honest feedback and respect your opinions in your review. We have redoubled our efforts to improve this paper because we sincerely believe the integration of traditional electromagnetic sensor based remote sensing with big data like social media based social sensing is a novel concept which will become increasingly relevant to remote sensing science, particularly in the context of environmental justice. We truly appreciate the time you have dedicated to reading and critiquing our work and hope that the revisions we detail below make our manuscript worthy of further consideration.
However, the fundamental problem at this stage is the diverse and complex analysis, only a small part of which relates to remote sensing. Moreover, there is no clear answer to the research questions. These ambitious and over-generalized research questions should be tightened.
Response: Thank you for this comment. We have sought to clarify our analysis and its relevance to remote sensing by reframing the way in which the study is presented in our introduction and reviewed in our discussion and conclusion. Our intention is to recognize the role that remote sensing plays in environmental justice research today and simultaneously acknowledge the opportunity to incorporate a human component through the integration of socially sensed big data. Additionally, to further accentuate our intention of testing this application, we have added a spectral index selection component to this study. In this process, we test four vegetation indexes and two water indexes to select the ones which best fit green and blue space measures for our model. Regarding our research questions, we have adjusted the wording and numbered each so we may refer back to them in the Discussion section. Our intention is for our linear models to help answer research question 1 (is there a relationship between variables), our new index selection process to help answer question 2 (how is this best modeled for remote sensing imagery and social media data), and our critical discussion on the measures and subsequent wildfire case study to help answer question 3 (what are the potential drawbacks). We hope that these adjustments will help to better convey our intended contribution to remote sensing science.
I suspect that there is good research paper (or two) that could be generated from the data. The paper needs to be reduced in length by about half. Once rewritten the authors may wish to consider submitting to a geography or planning journal. As applied research, the paper should be addressed to decision-makers, with recommended courses of action.
Response: Thank you for your comment. We agree the manuscript was becoming particularly lengthy, an issue which contributed to the feeling of disorganization. While we were unable to reduce the length of the paper by half, we have cut a significant amount of content while maintaining information we deem most relevant, reducing the length of the paper from 26 pages to 21. We hope this change helps to clarify the intention of our study and its applicability.
If the authors wish to produce something specifically relating to remote sensing I suggest looking at different ways of determining green and blue space, other than NDVI. They acknowledge weaknesses of the NDVI approach and state that conventional approaches use NDVI, so are there better alternatives? They could also test their belief that a single scene or mosaic is sufficient.
Response: We appreciate this insightful comment. We have followed your advice and included six total indexes (4 vegetation and 2 water body), including the NDVI and MNDWI originally present, in order to select the best one for our model. While the MNDWI was still the most appropriate water body index, the performance of the NDVI was surpassed by the NDMI by a small margin. This result was interesting and served to further our discussion on the relationship between remote sensing and social media in the context of environmental justice. These changes are present in the Data section in particular.
My recommendation is that the paper be rejected, and that there be no further encouragement to resubmit.
Response: Thank you again for your review. We believe the comments you have made have helped us to drastically improve our manuscript. While we do respect your opinions, we also argue that a refreshed look at remote sensing per the special issue’s attempt to integrate big data-based social sensing into the grand picture of remote sensing science is of great benefit to the remote sensing research community.
Reviewer 2 Report
Comments and Suggestions for Authors
The paper demonstrates a serious and significant effort in disclosing the complex relationship between environmental justice awareness and socio-environmental factors using remote sensing, social sensing, and big data technologies. The authors carefully design the methodology by extensively reviewing relevant factors, elaborately selecting variables, data collection methods, as well as analysis models. The paper thoroughly analyzes and discusses findings, particularly, the negative relationship between the number of tweens containing environmental justice words and ecosystem service measured by greenery, and also the negative relationship between the number of tweets and social vulnerability variables. It also insightfully analyzes and discusses limitations and issues of using social sensing for environmental justice awareness with the Tweeter device. Overall, the paper will make a significant contribution to the subject area and provide insights into public sentiment understanding for environmental policymakers.
Possible typo: Page 16, Line 698: SAR Error Model Results – “SEM”(?) Error Model Results
Author Response
The paper demonstrates a serious and significant effort in disclosing the complex relationship between environmental justice awareness and socio-environmental factors using remote sensing, social sensing, and big data technologies. The authors carefully design the methodology by extensively reviewing relevant factors, elaborately selecting variables, data collection methods, as well as analysis models. The paper thoroughly analyzes and discusses findings, particularly, the negative relationship between the number of tweens containing environmental justice words and ecosystem service measured by greenery, and also the negative relationship between the number of tweets and social vulnerability variables. It also insightfully analyzes and discusses limitations and issues of using social sensing for environmental justice awareness with the Tweeter device. Overall, the paper will make a significant contribution to the subject area and provide insights into public sentiment understanding for environmental policymakers.
Possible typo: Page 16, Line 698: SAR Error Model Results – “SEM”(?) Error Model Results
Response: Thank you very much for reviewing our study, we greatly appreciate your kind words. You are correct about the typo on line 698. We have corrected the error and thank you for pointing it out!
Reviewer 3 Report
Comments and Suggestions for Authors
This study suggests that the integration of big data, including Twitter activity and remote sensing imagery, offers a promising avenue for identifying and addressing environmental injustice. It emphasizes the potential for these technologies to enhance our understanding of environmental justice in both theory and practice, providing a foundation for future research and policy development. However, there are some issues that need to be resolved before this manuscript can be considered for publication. My recommendation is minor revision. More specific comments can be found in the attachment.
1. The scientific novelty from a remote sensing perspective is not clear. This is a remote sensing journal, so you should highlight the contributions of your study considering the subjects.
2. The introduction needs to be simplified.
3. Please provide a bit more big-picture motivation of how your analyses benefit society and how they have evolved over the past decade. However, from my point of view, the article does not provide a sufficiently thorough review of the issue under study. There are good references for the study techniques, but the paper is missing a "big-picture" introduction with some references in my opinion. I suggest that the authors should do a better analysis of the literature. It seems that the bulk of the text is a sort of compilation of statements in the individual articles cited. It would be better, I think, to extract ideas from individual articles and tie them together into a more fluid and conceptually homogeneous text. As it is, the text looks rather clumsy.
4. Research gaps, objectives of the proposed work should be clearly justified before the problem formulation section. This paper includes some little useful information and the main objectives of the study is not well defined. Problem statement is not clear and the objectives are obscure. Furthermore, the paper lacks a very clear and good justification for what is new and innovative about this case or this approach.
5. A research flowchart can help readers understand your study.

Comments on the Quality of English LanguageMinor editing of English language required
Author Response
This study suggests that the integration of big data, including Twitter activity and remote sensing imagery, offers a promising avenue for identifying and addressing environmental injustice. It emphasizes the potential for these technologies to enhance our understanding of environmental justice in both theory and practice, providing a foundation for future research and policy development. However, there are some issues that need to be resolved before this manuscript can be considered for publication. My recommendation is minor revision. More specific comments can be found in the attachment.
Response: Thank you for taking the time to review our manuscript. We will work diligently to address each of your concerns outlined below.
- The scientific novelty from a remote sensing perspective is not clear. This is a remote sensing journal, so you should highlight the contributions of your study considering the subjects.
Response: Thank you for this comment. We recognize the contribution our study intends to make to remote sensing science was diluted following previous revisions. To correct this, we have reduced the overall length of the paper and pushed remote sensing discussions to the forefront. In particular, this involved an extensive revision of the introduction, wherein the contribution of remote sensing to environmental justice research is acknowledged before the potential integration with social sensing big data is highlighted as the subject of the study. Similarly, the discussion section and conclusion have been revised to further place the study in the context of remote sensing. Finally, an additional remote sensing component was included in the form of a spectral index selection process. Rather than utilize the most popular indexes, we review several different vegetation and water indexes and select the ones which best fit our model. This addition can be found in the Data section, specifically 2.2.2.1.
- The introduction needs to be simplified.
Response: Thank you for this comment. As mentioned above, we have significantly reduced the overall length of the paper and, per your suggestion, much of this process focused on the introduction section. We have sought to reduce redundancy across the paper and have simplified some of the background discussions present in the introduction. We hope that these revisions have reduced the unnecessary complexity.
- Please provide a bit more big-picture motivation of how your analyses benefit society and how they have evolved over the past decade. However, from my point of view, the article does not provide a sufficiently thorough review of the issue under study. There are good references for the study techniques, but the paper is missing a "big-picture" introduction with some references in my opinion. I suggest that the authors should do a better analysis of the literature. It seems that the bulk of the text is a sort of compilation of statements in the individual articles cited. It would be better, I think, to extract ideas from individual articles and tie them together into a more fluid and conceptually homogeneous text. As it is, the text looks rather clumsy.
Response: Thank you for this insightful comment. You are correct in that we did not previously introduce the relevance of this study within a larger context, missing an opportunity to pinpoint the potential importance to society at large. To correct this and make our text flow more smoothly, we have reorganized the introduction, discussion, and conclusion sections. In particular, we hope to paint a bigger picture with our opening acknowledgement of how environmental justice identification tools today can results in prolonged harm due to their reliance on static data. We provide an example of lead contamination in Newark, New Jersey, wherein residents were experiencing lead poisoning which was not detected by existing environmental justice tools, pointing to a need for updated early detection tools.
- Research gaps, objectives of the proposed work should be clearly justified before the problem formulation section. This paper includes some little useful information and the main objectives of the study is not well defined. Problem statement is not clear and the objectives are obscure. Furthermore, the paper lacks a very clear and good justification for what is new and innovative about this case or this approach.
Response: Thank you for your comment. We recognize the research gaps, justifications, and objectives for our study were not clearly stated in our original manuscript. We have sought to address each of these issues in our revamped introduction section. This includes a statement on the need to explore the research gap present regarding the intersection of remote sensing imagery and social sensing big data, the inclusion of a big picture introduction, and a reframing of the research questions. We hope these changes will help to clarify the importance of this study and the contribution it aims to make to remote sensing science.
- A research flowchart can help readers understand your study.
Response: Thank you for this suggestion. While we recognize a flowchart would further clarify the structure of our study, we are concerned about our manuscript’s length. As pointed out by another reviewer, the extensive nature of the paper diluted our study and made it less approachable, and so we are forced to be conservative with space in this revision. We acknowledge that the previous version of our introduction presented an unclear framework for our study, and we hope the restructuring we have done has still helped to clarify the overall flow.
The below comments were present in the attachment provided by the reviewer. Line numbers below refer to that document, not the updated manuscript. We thank the reviewer for each of these comments and corrections and have outlined our attempts to address them below.
Line 42: In the introduction part, you only mentioned US. In fact, a big-picture background should be included in this section.
Response: The statement regarding the study’s importance has been broadened and an example application has been added.
Line 232: You should explain the reason.
Response: We have added additional statements highlighting the reason for selecting our study region.
Line 264: This section is too redundant. The authors should cut some description.
Response: We have sought to remove redundancies and repetition, reducing the overall length of this section.
Line 375: the reflectance of NIR band
Response: This particular instance has been removed, but we have attempted to ensure this mistake is not made in other places.
Line 375: Only in theory
Response: This is true, this statement has been removed in place of a broader discussion on spectral indexes.
Line 412: Serial number of Eqs should be added.
Response: Labelling has been added.
Line 562: additional .
Response: Removed, thank you for catching this error.
Table 2: sub
Response: These scripts have been adjusted.
Line 592: extra-
Response: Removed, thank you.
Line 696: the meaning of *, **, and *** should be added.
Response: An additional note has been added with this information.
Line 1059: Too long to read
Response: We have significantly reduced the length of the conclusion and recentered the conversation overall.